# Plant Alkylbenzenes and Terpenoids in the Form of Cyclodextrin Inclusion Complexes as Antibacterial Agents and Levofloxacin Synergists

**DOI:** 10.3390/ph15070861

**Published:** 2022-07-14

**Authors:** Igor D. Zlotnikov, Natalya G. Belogurova, Sergey S. Krylov, Marina N. Semenova, Victor V. Semenov, Elena V. Kudryashova

**Affiliations:** 1Department of Chemistry, M.V. Lomonosov Moscow State University, Leninskie Gory, 1/11B, 119991 Moscow, Russia; zlotnikovid@my.msu.ru (I.D.Z.); nbelog@mail.ru (N.G.B.); 2N. D. Zelinsky Institute of Organic Chemistry RAS, 47 Leninsky Prospect, 119991 Moscow, Russia; forvard1953@yandex.ru (S.S.K.); vs@chemical-block.com (V.V.S.); 3N. K. Koltzov Institute of Developmental Biology RAS, 26 Vavilov Street, 119334 Moscow, Russia; ms@chemical-block.com

**Keywords:** allylpolyalkoxybenzenes, terpenoids, synergism, levofloxacin, cyclodextrin inclusion complex, eugenol, apiol

## Abstract

Allylpolyalkoxybenzenes (APABs) and terpenoids from plant essential oils exhibit a range of remarkable biological effects, including analgesic, antibacterial, anti-inflammatory, antioxidant, and others. Synergistic activity with antibiotics of different classes has been reported, with inhibition of P-glycoprotein and impairment of bacterial cell membrane claimed as probable mechanisms. Clearly, a more detailed understanding of APABs’ biological activity could help in the development of improved therapeutic options for a range of diseases. However, APABs’ poor solubility in water solutions has been a limiting factor for such research. Here, we found that complex formation with β-cyclodextrins (CD) is an efficient way to transform the APABs into a water-soluble form. Using a combination of spectroscopic (FTIR, NMR, UV) methods, we have estimated the binding constants, loading capacity, and the functional groups of both APABs and monoterpenes involved in complex formation with CD: ethylene, aromatic, methoxy and hydroxy groups. In the presence of a molar excess of CD (up to 5 fold) it was possible to achieve the complete dissolution of APABs and terpenoids in an aqueous medium (at 90–98% encapsulation) higher by 10–1000 times. Further, we have demonstrated that CD-APABs, if used in combination with levofloxacin (Lev), can be antagonistic, indifferent, additive, or synergistic, mostly depending on the concentration ratio: at high Lev concentration with the addition of APAB is typically neutral or even antagonistic; while at a Lev concentration below MIC, the addition of CD-APAB is either additive or synergistic (according to FICI criteria). An over three-fold increase in Lev antibacterial activity was observed in combination with eugenol (EG), as per the growth inhibition diameter measurement in agar. Interestingly, a synergistic effect could be observed with both Gram-positive and Gram-negative bacteria. So, obviously, the APAB-CD and terpenoid-CD mechanism of action is not limited to their interaction with the bacterial membrane, which has been shown earlier for CDs. Further research may open new prospects for the development of adjuvants to improve the therapeutic regimens with existing, as well as with new anti-infective drugs.

## 1. Introduction

Plant extracts from dill *Anethum graveolens* L. (Apiaceae), parsley *Petroselinum sativum* Hoffm. (Apiaceae), nutmeg *Myristica fragrans* Houtt. (Myristicaceae), garlic *Allium sativum* L. (Amaryllidaceae), cloves *Syzygium aromaticum* L. (Myrtaceae) are aromatic oils that contain biologically active terpenes and polyphenols with analgesic, antibacterial, anti-inflammatory properties and even an immunomodulatory effect [1,2]. Plant essential oils are relatively safe for human and animal health and the environment, which supports their potential use in medicine as independent components, and in combined forms to improve the pharmacological properties of existing drugs.

A relevant representative of fragrant substances promising for medicine is eugenol (EG, 4-allyl-2-methoxyphenol). EG is an oily liquid, which is poorly soluble in water [3,4]. EG demonstrates antioxidant activity (inhibits lipid peroxidation), and also promotes the removal of free radicals [5,6,7]. It is known that EG in concentrations of several mg/mL has an antibacterial effect on some on Gram-positive (*Bacillus cereus*; *Bacillus subtilis*; *Staphylococcus aureus*) and Gram-negative (*Escherichia coli*; *Salmonella typhi*; *Pseudomonas aeruginosa*) bacteria [8,9]. Hemaiswarya et al. [10] showed that EG acts as a synergist of the antibacterial activity of several antibiotics (ampicillin, tetracycline, rifampicin, chloramphenicol) against Gram-negative bacteria. The addition of EG as an adjuvant can reduce the MIC of antibiotics by 5–1000 times (the highest values are characteristic of tetracycline and vancomycin). It is suggested that EG forms defects in the membranes of bacteria at a concentration of 1 mM. The second aspect of the action is the inhibition of mitochondrial dehydrogenases [11]. As promising adjuvants, apiol and its analogs dillapiol, myristicin, nothoapiol should be noted [1,12]. Some monoterpenes (-)-carveol, geraniol and citronellol, show a synergy effect when used as additives for conventional antimicrobials (chloramphenicol, minocycline, amoxicillin and ciprofloxacin) [13,14].

However, due to their poor aqueous solubility, volatility, oily consistency, tendency to oxidation of EG and other APABs, investigation of their biological activity, including synergy with antibiotics, has been limited. Various methods have been tried to tackle this problem, including nanospheres and lipid nanoparticles. EG-loaded chitosan nanospheres (760 kDa, diameter 80–100 nm) [14] showed 12.8% loading capacity (LC) of EG by weight, and an encapsulation efficiency of 16%. Similar loading parameters were obtained for lipid carriers with loaded EG; however, for practical applications, this is deemed sub-optimal [15].

Much higher loading parameters were reported for cyclodextrins (CDs), forming inclusion complexes with poorly soluble aromatic or hydrophobic substances and drugs [16]. CDs are cyclic oligomers of glucopyranose residues connected by α(1→4)-glycoside bonds, representing a truncated torus with an external hydrophilic shell and an internal hydrophobic cavity. Their complex formation improves solubility, protects from oxidation, degradation, hydrolysis, enzymatic destruction [17], increases penetration through biological membranes [18], and may increase circulation time in the bloodstream. EG was shown to solubilize in CDs, forming energetically advantageous complexes with β-CDs (according to modeling data, total free energy is about −85 kcal/mol), in particular, with 2-hydroxypropyl-cyclodextrin (HPCD [19,20,21,22]. The β-form of CD, in comparison with α and γ, is characterized by higher values of complexation constants (1000 M^−1^ vs. 10–100 M^−1^), which allow the efficiency of EG loading (and approximating for other aromatic molecules) to be increased and a prolonged release: the half-dissociation period is 3 h versus 1 h for EG-αCD [22]. EG was found to inhibit the growth of *E. coli* only in the form of a complex with CD (concentration 0.5 mg/mL is already active), but not in the free form. Almost 92% of the initial amount of EG could be included in CD (host:guest molar ratio of 1:1), while the mass fraction of EG in the sample was about 58% [23]. Available data indicates that CD complex formation may be superior to other loading systems, including nanoparticles and liposomes, in terms of loading capacity LC (>60%), entrapment efficiency (EE > 90–95%), stability of complexes and simplicity of preparation [24,25,26].

However, structural information about such complexes is lacking. We have studied the APAB and monoterpene complex formation with β-cyclodextrins using a combination of spectroscopic methods (UV, FTIR and NMR) to determine the binding constants, the functional groups involved in such binding. Furthermore, as synergistic effects with antibiotics were reported for pure EG (without carrier), we aimed to study the effect of CD complex formation on similar biological activity for a series of isomeric APABs and monoterpenes. Levofloxacin was selected as a broad-range antibiotic, acting on Gram-negative and Gram-positive bacteria. Some essential oil components were also reported as Levofloxacin (Lev) enhancers [25,26].

## 2. Results

### 2.1. Determination of Inclusion Complex Formation between HPCD and Eugenol, Apiol

Natural allylpolyalkoxybenzenes extracted from the seeds of crop plants of the family Apiaceae, dill *Anethum graveolens* L., parsley *Petroselinum sativum* Hoffm., and fennel *Foeniculum vulgare* Mill., exhibited a wide range of biological potencies, including antiproliferative, anti-inflammatory, antimicrobial and antifungal activities. However, poor solubility of these compounds (EG, apiol and others) significantly limits their applicability. The formation of inclusion complexes with CDs suggested here was used to increase the aqueous solubility of these lipophilic compounds. The selection of the optimal parameters for the loading of drugs into CDs would allow new formulations of therapeutic drugs to be obtained with a high proportion of the target agent with increased activity.

#### 2.1.1. UV Spectroscopy Approach

The formation of inclusion complexes of terpenoid’s and APAB’s with MCD was investigated by UV spectroscopy (Figure 1). In the UV spectra of EG (Figure 1A), safrole (Figure 1B) and its MCD-complexes, three peaks were observed: (1) 279 and 287 nm, respectively, correspond to the absorption of the aromatic system; (2) 205 nm—absorption of the double bond of the allyl group; (3) 230 and 235 nm, respectively, correspond to the resonant structures of a double bond with a benzene ring. The region from 240–350 nm was analyzed for all compounds studied except for linalool absorbing due to two double bonds at 200–220 (ε_200_ = 24,300). For pure APAB, the molar absorption coefficient was in the order of thousands (Figure 1C), in the series of apiol’s analogues increased from 1580 for myristicin to 4570 for allyltetramethoxybenzene symbiotically with an increase in the number of CH_3_O groups and their spatial convergence (dillapiol vs. apiol).

The formation of APAB inclusion complexes with MCD led to an increase in the UV absorbance of the samples at 278–288 nm (Figure 1), which corresponded to an increase in the concentration of these substances in aqueous fraction (APAB_(aq.)_ alone and APAB-MCD_(aq.)_). In Figure 1A,B, the brown spectra correspond to EG and safrole dissolved in water without MCD, while the cyan spectra represent a system APAB–MCD (molar ratio 1:5) with increased solubility of APAB by 5 and 25 times, respectively, for EG and safrole.

To optimize the system composition, similar experiments were carried out on the formation of EG and apiol inclusion complexes with HPCD (Appendix A). In addition to changes in intensity, the following changes were observed in the UV spectra of apiol and EG: the peaks of apiol and eugenol shift of 205 nm into the long-wavelength region occurs due to a decrease in the energy of the π→π*, n→π* allyl group transitions into the hydrophobic cavity of the CD compared to aqueous solution. According to these changes, the dissociation constants of the EG–HPCD and apiol–HPCD complexes (Appendix A) were calculated by Hill’s equation (Table 1). Similar changes were observed in the complexation of EG and apiol with MCD. The physicochemical parameters of the interaction of APAB and terpenoids (X) with CD were studied by determining the concentration of substances in the aqueous fraction ([APAB] + [APAB-MCD]) according to the calibration dependence (Figure 1C) for systems with a constant amount of the essential oil component X and the concentration gradient MCD (Table 1). Theta (θ) is a fraction of the bound substance X:θ = [X-MCD]/([X-MCD] + [X]) and was calculated based on the dependence of the absorption intensity X on the MCD concentration in the system in the form of a sigmoidal Hill curve with subsequent linearization (Section 3, Section 3.3 and Section 3.4).

APAB and terpenoids form inclusion complexes with CD with an excess of tori per guest molecule from 1.1 to 1.7, which is presumably explained by the smaller volume of the included molecule in comparison with the CD cavity, as well as the possibility of interaction additionally with the outer hydrophilic shell of the CD. The distinctive properties were demonstrated by myristicin, which forms inclusion complexes of molar ratio APAB:CD close to 2 to 1. Typical values of K_d_ are of the order of magnitude 10^−^^3^–10^−^^4^ M. In general, MCD forms inclusion complexes stronger by 0.5–1.5 orders of magnitude than HPCD due to the increased hydrophobicity of the inner cavity due to methyl groups [27]. The inclusion complex is stronger in the case of larger and more hydrophobic guest molecules. Safrole, myristicin and allyltetramethoxybenzene are the leaders among the APABs. The quantitative loading of drugs into the CD can be estimated using the parameters entrapment efficiency (EE) and loading capacity (LC). EE characterizes the proportion of molecules that form a complex with CD. In the presence of 0.01 M MCD, almost all of the considered compounds effectively form complexes with CD, EE > 40%, although this parameter is lower for apiol. At the same time, Lev, menthol, myristicin and safrole are more than 90% in a complex form at the studied conditions. LC characterizes the mass fraction of the “guest” in the complex formation. In the considered series, LC ≈ 10%, increases for compounds with a higher molecular weight and stronger binding to CD.

The graphical dependence of EE (calculated based on the absorption at 280 nm and peak shift at 205 nm in the UV spectra) on the amount of HPCD is given in the Appendix A. The entropy factor is the driving force behind the formation of APAB-CD complexes [22]. The dissociation of EG–HPCD and apiol–HPCD complexes is kinetically impeded: the amount of the complex remains at a level above 95% of the initial at 100-fold dilution (from 10 mM to 0.1 mM—relevant concentrations during biological testing) at 22–37 °C. The formation of APAB and terpenoid inclusion complexes is important for a significant increase in the solubility of these compounds in water (Table 1). A sample of 2 and, more effectively, 10 mM CD enables an increase in the solubility of hydrophobic compounds by several times, which allows them to be used as the biologically active adjuvants of antibacterial drugs. In fact, solubility is limited by the concentration of CD, since the EE for many of the compounds under consideration is >90%, which means that almost any reasonable concentration of the complex form is achievable.

#### 2.1.2. FTIR Spectroscopy Approach

Fourier transformed infrared spectroscopy in the mode of attenuated total reflection (ATR) is an advanced method for studying the interactions of substances at the atomic-molecular level. Changes in the microenvironment of chemical bonds, functional groups, protonation/deprotonation, or the formation of non-valent interactions between the receptor and the ligand (in this case, the formation of guest–host inclusion complexes of CD with EG and apiol molecules) leads to a change in the intensity or position of characteristic peaks in the IR spectra. Thus, when the apiol was included in HPCD or MCD (data for MCD are similar and not shown), a significant increase in the intensity of all characteristic peaks was observed (Figure 2A): 1652 cm^−^^1^—ethylene ν(C=C), 1608 and 1505 cm^−^^1^—aromatic ν(C–C), 1454 cm^−^^1^—methylene δ_s_(C–H), 1430 and 1353—methyl δ_s_ and δ_as_ (C–H), overlapping peaks ν(C–O), ν(C–O–C), δ(O–H) bond oscillations were observed in the region of smaller wave numbers for 1300 cm^−^^1^ [19]. This fact indicates the inclusion of apiol not only in the CD cavity, but also its interaction with the external hydrophilic shell of HPCD. The formation of inclusion complexes of APAB with a CD cavity was confirmed with a change in the intensity of peaks in the IR spectra, first of all corresponding to the fluctuations of aromatic C-C bonds (1505 cm^−^^1^) and ethylene C=C (1650 cm^−^^1^). These data allowed the estimation of the apiol–CD complex dissociation constant value as 4.0 ± 0.3 mM (Appendix A). In this case, one CD molecule interacts with two apiol molecules, (by Hill linearization model). The above-indicated peaks in the IR spectra are composite which was visually confirmed by the changes in the shape of peaks and even the appearance of shoulders when the APAB was turned on in the CD cavity. To separate the complex peaks into components, increase sensitivity and obtain more detailed information, the differentiation method was used. The shifts of minima/maxima of the second derivative of spectra (Figure 2B) in allyl ν(C=C) and aromatic ν(C–C) regions were the most prominent. At the same time, when the apiol was included in CD, similar, but less vivid shifts of peaks corresponding to the C-H, C-O bonds occurred, which indicated the inclusion of the entire APAB molecule into the CD cavity.

The interaction of EG with CD leads to opposite changes in the IR spectra: quenching of almost all characteristic peaks was observed (Figure 2C). This can be explained by the possibility of apiol interacting with the outer hydrophilic shell of the CD (in addition to the CD cavity itself), which is less typical for EG, which forms an order of magnitude stronger guest–host inclusion complex. Moreover, this phenomenon depends on the amount of CD added, as we have previously demonstrated by the interaction of EG with mannosylated polymer and molecular containers [29]. Nuchuchua et al. [22] have also observed a decrease in the intensity of the EG peak in the solid-phase EG–CD spectra. In addition, there are examples of the “disappearance of guest molecules” in the CD cavity [30] or peak’s quenching [31,32,33], especially for model system CD–fluoroquinolone [34]. However, at the same time, a bifurcated peak appears and increases in intensity (1738, 1730 cm^−^^1^)—which corresponds to the overtone of deformation vibrations of C–H bonds of the terminal methylene group. Secondly, the aromatic peak of EG 1514 cm^−^^1^ shifts to 1515.5 cm^−^^1^ (Appendix A), which made it possible to calculate the dissociation constant of the EG–HPCD complex equal to (4.1 ± 0.3) mM. The graphs of the second derivative (Appendix A) provided a visual representation of the change in the microenvironment of functional groups and the aromatic system of the molecule included in the CD cavity. These parameters were in a good agreement with the UV spectroscopy data as well as the shift of the peak at 205 nm to the long-wavelength region in the UV spectrum indicating that EG was included in HPCD cavity.

To level the overlap of bond fluctuations in water molecules and the effect of hydration, the spectra of the EG–HPCD complex in solid form were registered as complementary materials (Appendix A). There are characteristic peaks of EG, which were quenched in the aqueous phase, C–C_arom_ and C=C of the ethylene group. In the solid phase, we observed peaks corresponding to the stretching oscillations of CH_2_, CH_3_ groups (2900–3000 cm^−^^1^) and O-H (3200–3500 cm^−^^1^) without solvent contribution. Appendix A shows the distribution (significant for visualization and semi-quantitative determination of drug loading parameters EE, LC) of the integral intensity of the C–C_arom_ peak, which characterizes the allocation of eugenol in the sample. EG was effectively (EE > 70%) included in two-thirds of the sample by quantity, in the remaining part EE exceeded 25–35%. 

Visually, the dissolution of APAB and the homogenization of the system (disappearance of oil droplets) when they were included into CD (both in the MCD and in the HPCD) were observed in a light microscope (Appendix A). The dependence of the dissolution degree on the content of CD in the system was determined. Visual observation indicated that for a noticeable conversion to the complex/soluble form of EG, an equimolar amount of CD is required, and almost complete dissolution is achieved with a 5-fold molar excess. For apiol, due to its lower affinity for CD, 2- and 10–15-fold excess of CD are required, respectively. The degree of EG inclusion practically does not increase with long incubation. Since soluble complex forms of EG, apiol and other individual components of essential oils have been obtained, a quantitative study of the adjuvant and synergistic activities of APAB is relevant but previously impossible because of the stratification of substances and the formation of oil fractions. The following are the results of experiments to determine the antibacterial and synergistic effects of adjuvant–CD complexes.

#### 2.1.3. NMR Spectroscopy

NMR spectroscopy is a useful tool, and complementary to FTIR, to provide the evidence for the inclusion of guest molecules in the CD cavity and interaction with the outer hydrophilic shell of the CD. The ^1^H NMR spectra of MCD and HPCD alone, their complexes with EG and apiol (2:1) as well as double inclusion complexes MCD-Lev-EG and HPCD-Lev-EG in D_2_O are presented in Figure 3 and Appendix A. The interactions of MCD and HPCD with EG, apiol and Lev and the structure of complexes can be characterized by induced chemical shifts Δδ, equal to the difference of chemical shifts in the complex and single substances (Table 2). The inclusion of an apolar fragment of the guest molecule into the host hydrophobic cavity induced a shielding of the inner protons of the glucose units of MCD, namely, H3 and H5, whereas the protons on the exterior of the torus (H1, H2 and H4) were relatively unaffected (Figure 3A—insert); this was previously shown in systems where CD formed inclusion complexes with Lev, atropine, nitrobenzene, nicardipine [35,36,37,38,39].

The effect of MCD on the chemical shift of EG protons (Figure 3B—insert), Lev protons (Figure 3C—insert) and apiol protons (Appendix A) is presented in Table 2. This effect is split into two groups: first chemical shifts are shifted upfield and the other downfield. A downfield chemical shift of the drug protons indicates that they are close to an electronegative atom, oxygen [38,40,41,42]. An upfield chemical shift is probably due to a change in the local polarity when the protons of the guest molecule are immersed in the MCD cavity (screening effect due to van der Waals forces between the drug and the carbohydrate chains of the CD).

**Table 2 pharmaceuticals-15-00861-t002:** ^1^H-chemical shifts corresponding to MCD and HPCD in the presence and absence of EG, apiol and mixture of EG-Lev. ^1^H-chemical shifts corresponding to EG and Lev in the presence and absence of MCD and HPCD in comparison with literature data for EG and Lev alone [35,41]. D_2_O, 600 MHz.

**MCD Proton Chemical Shifts (Insert of Figure 3A)**					
MCD proton	δ (MCD)	δ (MCD-EG)	δ (MCD-apiol)	δ (MCD-Lev-EG)	
H1	4.957	4.918	4.942	4.941	
H2 *	3.553	3.557	3.545	3.603 or 3.577 (overlapped)	
H3 *	3.886	3.843	3.868	3.852	
H4	3.454	3.472	3.454	3.492	
H5	3.763	3.736 (overlapped)	3.755	3.797	
H6 *	3.805	3.78	3.787	3.767	
**HPCD Proton Chemical Shifts (Insert of Figure 3E)**					
HPCD proton	δ (HPCD)	δ (HPCD-EG)	δ (HPCD-apiol)	δ (HPCD-Lev-EG)	
H1	5.279	5.369	5.303	5.287	
H1′	5.113	5.236	5.147	5.126	
H2 *	3.544	overlapped	3.593 (overlapped)	overlapped	
H3 *	3.908	3.918	3.948 (overlapped)	3.921	
H4	3.526	3.504	3.543 (overlapped)	3.568 (overlapped)	
H5	3.633	3.691	3.667	3.641	
H6 *	3.673		3.714	3.689	
CH_3_	1.193	1.331	1.225; 1.238	1.20; 1.212	
**EG Proton Chemical Shifts (Insert of Figure 3B)**					
EG proton	δ (free EG) [41]	δ (MCD-EG)	δ (HPCD-EG)	δ (MCD-Lev-EG)	δ (HPCD-Lev-EG)
Ha	3.305	3.406	overlapped	overlapped	3.359
Hb	3.871	3.829	3.765	3.853	3.857
Hc	5.034	4.993	overlapped	4.986–4.988	overlapped
Hd	5.47	5.527		overlapped	
He	5.902	5.887	6.175; 6.188	5.897–5.886	6.047; 6.059
Hf	6.683	6.611	6.903; 6.993; 6.979	6.614	6.763; 6.779
Hg	6.824	6.821	7.074; 7.081; 7.089	6.821	6.975; 6.995
**Lev Protons Chemical Shifts (Insert of Figure 3C)**					
Lev proton	δ (free Lev) [38]		δ (MCD-Lev-EG)	δ (HPCD-Lev-EG)	
H5	8.342		8.374	8.513	
H8	7.240		7.44	7.649	
H3′, H5′	3.384		3.381 (overlapped)	3.359 and 3.371 (overlapped)	
H2′, H6′	3.49		3.577–3.603 (overlapped)	3.5172	
H15	1.446		1.452	1.582 and 1.595	
H16	2.946		overlapped	2.952	
**Apiol Proton Chemical Shifts (Insert of Figure 3F)**					
Apiol proton	δ (free apiol) [12]		δ (MCD- apiol)	δ (HPCD- apiol)	
Ha	3.87		3.868	3.886	
Hb	3.84		3.787	3.817	
Hc	6.3		6.32	6.544	
Hd	5.94		overlapped	6.099	
He	3.31		3.28	3.433; 3.444	
Hf	5.95		overlapped	6.114	
Hg	5.09		5.15	5.147	

* Low intensity due to the predominant substitution of these protons by a methyl group in the MCD and hydroxypropyl group in the HPCD.

It is noteworthy that inclusion in the CD cavity is more pronounced for MCD, and to a lesser extent for HPCD having hydrophilic substituents, which causes induced chemical shifts in a weak field (Table 2; Figure 3E,F and Appendix A). Thus, an increase in the solubility of APAB and other terpenoids is achieved by including cyclodextrin in the cavity and interacting with the outer shell of the CD. The results obtained in the paper [27] show the role of methyl CD’s substituents that increase the hydrophobicity of the internal cavity, which affects chemical shifts: a shift in a strong field is characteristic only for MCD—therefore, the complexes formed are stronger and the solubility of APAB is higher. According to computer simulation data, the sizes of the hydrophobic cavities CD and MCD are practically the same and approximately equal to 10.2–10.3 Å (O6-O3) (Appendix A). At the same time, the HPCD cavity size is a bit smaller (8.9 Å) due to volumetric substituents. The height of the CD tori increases in the series CD < MCD < HPCD: 6.2, 6.6 and 8.6 Å (Appendix A). The simulation data were consistent with the literature data calculated based on X-ray diffraction (Appendix A) [43]. These values confirm the preference for the inclusion of hydrophobic molecules of the APAB type and terpenoids inside the MCD rather than HPCD, given the dimensional aspect and increased hydrophilicity of HPCD, which destabilizes the complex.

The formation of complexes with CD is additionally confirmed by the chemical shifts of protons of the MCD due to theimpact of the guest molecule (Lev/EG/apiol). The upfield shifts observed for the H3 (δ_EG_ > δ_EG-Lev_ > δ_apiol_), H5 and H6 (δ_EG_ > δ_apiol_ > δ_EG-Lev_) protons of MCD related to water replacement by the hydrophobic aromatic benzene rings of the EG, apiol and Lev molecules inside the cavity. The greatest induced shift is characteristic of a system with EG, which indicates in favor of a greater thermodynamic benefit of complexation. The downfield shifts of protons H2, H4 MCD for EG and Lev indicate additional interactions of polar groups (-OH, -COO^–^, C=O) with the outer hydrophilic shell of the CD, which increases the strength of the guest–host complex. In contrast, such stabilization is practically not observed for apiol, which explains the small values of K_d_ for apiol and higher values for EG and Lev determined by the UV and FTIR methods (Table 1).

Induced shifts of EG protons confirm this mechanism (Table 2). The downfield shifts of the Hd proton indicate the electrostatic interactions of the -OH group of the EG with the outer shell of the CD. The upfield shifts observed for the remaining protons confirms the inclusion of the aromatic system and the allyl group in the CD cavity. In the presence of Lev, there is competition from the guest molecules for CD, as a result of which the apparent constants of the formation of inclusion complexes decrease, which is reflected in a decrease in the magnitude of induced shifts. However, the δ(Hc) in the Lev-EG mixture is greater than for a simple EG, which is presumably due to additional π-π and π-p stacking between the allyl group of the EG and the aromatic Lev system directly inside the CD cavity. In other words, it can form a complex of inclusion of both molecules (Lev and EG) into one CD torus at once.

The interaction of Lev with MCD is characterized by the inclusion of a predominantly aromatic system inside the CD cavity, since the largest induced shifts (Table 2) are characteristic of the H5 and H8 quinolone structure. Moreover, the H14 signal is absent, which indicates that the carboxyl group is in a deprotonated state and its interaction with the outer hydrophilic shell. Induced proton shifts of the piperazine ring H2′ and H6′ are large, while H3′ and H5′ are hardly noticeable, which is explained by the location of half of the piperazine ring Lev outside the CD cavity (the molecule does not fit completely). Thus, the structure of the Lev-MCD complex is a complex of inclusion of the quinolone and half of the piperazine structure into the CD cavity and additional stabilization of the “emerging parts of the Lev” by the methyl and hydroxyl groups of the outer surface of the MCD.

The interaction of apiol with MCD leads to weak induced shifts due to low EE (Table 2). The upfield shifts observed for protons of methyl and methylene groups of apiol confirm the inclusion of MCD in the hydrophobic cavity. In the case of the interaction of apiol with HPCD, very strong downfield shifts are observed (Table 2), indicating the interaction of apiol with the hydrophilic outer shell of HPCD and the retention of molecules by hydroxypropyl groups.

The NMR spectroscopy data were in good agreement with those obtained by FTIR. Namely, both methods confirmed the inclusion of aromatic systems of APAB and Lev in the hydrophobic cavity of the CD; the possibility of including the carboxyl group of Lev in the CD only in the protonated state or as an alternative, the deprotonated COO^–^ interacts with the outer polar shell; the change in the microenvironment of the allyl group EG. Therefore, these methods are complementary and together allow us to study the mechanism of formation of the complexes.

Based on the above changes in the NMR spectra, we conclude that the most likely structure of the inclusion complexes EG, apiol and Lev is that shown in Figure 3D, which is in a good agreement with the data presented in the work [43].

Using spectroscopic methods, the formation of complexes of inclusion of aromatic and hydrophobic molecules with both MCD and HPCD was proved. The complex formation significantly increases solubility and protect functional groups responsible for antibacterial properties from destruction. Therefore, it is expected to increase the antibacterial efficiency of Lev, enhanced by APAB and terpenoids, in the form of complex formulations with CD.

### 2.2. Antibacterial and Synergistic Effects of APABs and Monoterpene Adjuvants

APAB adjuvants can be used to reduce the concentration of the potentially toxic Lev in the complex formulation due to the synergy effect, as well as to increase the therapeutic potential of the drug. Presumably, the use of complex forms of Lev in combination with APABs would increase the circulation time of Lev in the bloodstream and “save” a significant part of the drug from misuse (destruction, accelerated excretion). Indeed, EG has an antibacterial effect on some bacteria [8,9]; EG [10], apiol and its analogues [1,12] as well as monoterpenes, menthol and linalool [13], showed synergism with several antibiotics and antimicrobials (chloramphenicol, minocycline, amoxicillin and ciprofloxacin). In this paper, we studied the antibacterial effect of adjuvants (EG, apiol, dillapiol, myristicin, allyltetramethoxybenzene, linalool, menthol and safrole) in the form of inclusion complexes with MCD, as well as the effect of these formulations on the activity of Lev on Gram-positive and Gram-negative bacteria. The choice of Lev was based on its wide applicability in the treatment of a wide range of bacterial infections. In addition to APABs, two monoterpenes, linalool and menthol, were also studied as adjuvants since they were reported to exhibit antimicrobial effect [7,44,45]. A favorable adjuvant configuration was identified by comparison of compounds with different numbers and positions of substituents in the benzene ring (Table 3). 

#### 2.2.1. Antibacterial Effect of Lev, Lev–MCD and Adjuvants–MCD 

(1) *E. coli*. According to the agar diffusion test, Lev in its free form inhibited growth of *E. coli* with MIC = 0.1 μg/mL, whereas the application of the Lev inclusion complex in MCD reduced the MIC value to 0.06 μg/mL (Table 3). A similar tendency was observed according to the broth micro-dilution test, where MCD improved the effect of Lev by 30%. This phenomenon can be explained by defects in cell membranes arising due to CD, and as a consequence, an increase in the penetration of the Lev through the bacterial plasma membrane (as shown by Tychinina et al. [18,46] on the model experiment with liposome). The presented values of MIC50 and MIC90 determined for APAB and terpenoids in the form of inclusion complexes with MCD by broth micro-dilution technique were less in terms of diffusion in agar due to differences in the experiment conditions due to a slow diffusion rate in agar and a large bacterial seeding area. The values obtained by the two methods correlated with each other in terms of relative characteristics. Comparison of adjuvant MIC values yielded EG as the optimum molecular scaffold, namely, allylbenzene with 3-methoxy-4-hydroxy substituents. Safrole with a methylenedioxy moiety was second in the activity range, whereas apiol and dillapiol featuring a dimethoxy-methylenedioxy fragment were less potent (Table 3). The antimicrobial effect of methylenedioxy-containing APABs might be attributed to their facile hydrolysis in cells yielding molecules with a hydroxy-methoxy-substituted benzene ring [1]. CD complexed with myristicin, allyltetramethoxybenzene, and linalool were inactive on both bacteria species up to 3–5 mg/mL concentration. Thus, the ranked list of substances according to their activity against *E. coli* is the following: Lev-MCD > Lev >>> EG-MCD > dillapiol-MCD ≈ apiol-MCD > myristicin-MCD >> other (no significance).

(2) *B. subtilis*. According to the agar diffusion test, it was found that Lev and adjuvants inhibited growth of Gram-positive *B. subtilis* at higher concentrations than required for Gram-negative bacteria (Table 3, Appendix A). Menthol was the only exception. It failed to exhibit an effect on Gram-negative bacteria, while it suppressed the growth of Gram-positive *B. subtilis*, although at a relatively high concentration (MIC = 2.6 mg/L). Lev in its free form inhibited growth of *B. subtilis* with MIC = 0.45 μg/mL, whereas application of the Lev inclusion complex in MCD reduced the MIC value to 0.25 μg/mL. According to the broth micro-dilution method, MCD reduced MIC_50_ of Lev by half, and MIC_90_ by more than a third. Among the APAB and terpenoids studied, only EG (MIC = 1 mg/mL), apiol (MIC = 5.3 mg/mL), safrole (MIC = 3.9 mg/mL) and menthol (MIC = 2.6 mg/mL) showed an inhibitory effect on the growth of Gram-positive bacteria in reasonable concentrations. According to the broth micro-dilution method, safrole turned out to be more active than menthol, which is probably due to the slow diffusion of the latter in the agarose gel. The structure–activity relationship for adjuvants on Gram-positive and Gram-negative bacteria (considered for *E. coli*) was similar (Table 3). Thus, a ranked list of substances according to their activity against *B. subtilis* is the following: Lev-MCD > Lev >>> EG-MCD > safrole-MCD ≈ menthol-MCD > myristicin-MCD >> other (no significance).

#### 2.2.2. Synergy of Adjuvants with Lev

We obtained complex formulations containing adjuvants-MCD and Lev-MCD. These complexes were referred to as enhanced Lev. The synergy effect was studied at a constant concentration of Lev and a variable concentration of an enhancer. Table 4 shows the diameters (D_Lev_ and D) of *E. coli* and *B. subtilis* growth inhibition zones surrounding Lev-MCD and Lev–MCD with adjuvant–MCD, respectively. Based on the D/D_Lev_ ratio, the effectiveness of cooperative antibacterial action (ϕ) of the adjuvant and Lev was calculated. The fractional inhibitory concentration (FIC) parameter was introduced: at a fixed Lev concentration, the FIC of adjuvant corresponds to the minimum fractional adjuvant concentration at which the antibacterial effect was still manifest (MIC_adjuvant_ in the presence of Lev) [47]. The fractional inhibitory concentration index (FICI) [27] characterizes the ratio of the active Lev and adjuvant concentrations to the MIC values of these substances individually:$$\mathrm{FICI}=\frac{\left[\mathrm{Lev}\right]}{{\mathrm{MIC}}_{\left(\mathrm{Lev}\right)}}\times \frac{\left[\mathrm{adjuvant}\right]}{{\mathrm{MIC}}_{\left(\mathrm{adjuvant}\right)}}$$
(see Section 3). 

The FICI was interpreted as: synergism (<0.35), additivity (0.35–0.75), indifference (0.75–1.4), or antagonism (>1.4). FICI is designed to determine the type of Lev interaction with adjuvants. The important parameter shows the decrease in MIC of Lev due to the presence of an adjuvant: [Lev]/MIC_Lev_ = FICI/FIC_adjuvant_.

As discussed above [13], the addition of APAB-CD and terpenoid-CD complexes to an antibacterial drug reduced its MIC, resulting in the drug dosage decrease and, consequently, reducing the systemic toxicity and development of drug-resistant pathogens.

(1) *E. coli*. Significant enhancement of Lev inhibitory effect corresponded to a decrease of MIC_Lev_: up to twice in the presence of menthol (Appendix A) and apiol (Appendix A), up to 1.25–1.5 times when dillapiol, myristicin and allyltetramethoxybenzene were used as adjuvants. The most powerful effect was shown for safrole and EG (Figure 4a): MIC_Lev_ can be reduced from 0.10 to 0.015–0.02 μg/mL. 

The greatest increase in antibacterial power ϕ was demonstrated using the following combinations (Table 4): (a) EG-MCD + Lev-MCD (EG:Lev = 10^4^:1), ϕ = 3.3; (b) apiol-MCD + Lev-MCD (apiol:Lev = 5 × 10^4^:1), ϕ = 2; (c) safrole-MCD + Lev-MCD (safrole:Lev = 3 × 10^4^:1), ϕ = 2; and (d) menthol-MCD + Lev-MCD (menthol:Lev = 3 × 10^4^:1), ϕ = 2 (Table 3). The adjuvant effect correlates with the individual antibacterial activity of APABs, while the resulting enhancement of the Lev effect is not limited to a simple sum of activities, but is characterized by significant synergism.

FICI values characterize the type of Lev interaction with APABs or terpenoid. Almost all combined forms studied were characterized by an enhancing of the effect compared to the action of individual components. The additivity type already indicated a good potential of a pair of components, and the synergism type indicated a significant decrease in the fractional concentration of one of the components. The desired reduction of the MIC of potentially toxic Lev was demonstrated by EG, apiol, dillapiol, myristicin, menthol and safrole. The most powerful synergist, EG, exhibited maximal values of the two essential parameters, ϕ and FICI, at the relevant Lev concentrations of 0.5–2 MIC. However, the increase in Lev concentration up to >8–10 MIC resulted in indifference or even antagonism due to a saturation effect.

The graphical dependences of the *E. coli* growth inhibition zone diameters by Lev at varying adjuvant concentrations (from 60 μg/mL to 1–3.3 mg/mL) are shown in Figure 5. The greatest synergistic effect corresponded to safrole, apiol and menthol (Figure 5, Table 4). In the first experiment (Figure 4a), the Lev concentrations were selected in such a way that Lev concentration without an adjuvant was close to the MIC. In the second experiment (Figure 5b), inhibition by Lev itself was not achieved; however, the addition of adjuvants resulted in significant increase inhibition zone diameters from 10 to 13–14.5 mm.

The strongest synergist of levofloxacin among the APAB and terpenoids was eugenol (Table 4). Clearly and in detail this effect is presented in the form of 3D dependence of the inhibitory effect of the drug composition on the concentration of Lev and EG (Figure 6a): yellow area—weak inhibition, orange—medium, red—significant inhibition. MIC of Lev-MCD on *E. coli* was reduced from 0.06 to 0.015–0.02 μg/mL due to introduction of 1 mg/mL of the adjuvant EG-MCD. At minimal Lev concentrations the greatest adjuvant effect of EG (a 3-fold amplification effect) is achieved due to synergism. 

(2) Bacillus subtilis. The synergistic effect of EG-MCD (Appendix A), safrole-MCD (Appendix A) and menthol-MCD (Figure 4b) with Lev-MCD was also confirmed on Gram-positive bacteria (Table 4). All three adjuvants demonstrated an increase in antibacterial activity with an efficiency parameter ϕ from 1.1 to 1.5; the most powerful effect was observed for menthol: with ϕ value of 2.4. Moreover, the type of interaction of safrole and menthol with Lev was a «bright synergy» (FICI—0.08–0.11). However, in the case of Gram-positive cells, eugenol showed only additivity. Figure 6b shows the 3D dependence of the inhibitory effect of the drug composition on Lev and menthol concentrations: purple area—weak inhibition, blue and green—medium, and yellow-red—significant inhibition. Menthol-MCD at 1–3 mg/mL was identified as the strongest enhancer of Lev. MIC_Lev_ reduced from 0.45 to 0.15 μg/mL. However, EG-MCD was the strongest *B. subtilis* growth inhibitor itself. Presumably, the antibacterial properties of eugenol are due to its ability to destroy cell membranes; in contrast, the adjuvant power of menthol is explained by the predominant intracellular action. Facilitating drug permeability through bacterial membrane is a crucial factor for an antibacterial effect on Gram-negative bacteria. As an example, EG demonstrated the greatest antibacterial activity along with pronounced membrane destruction, which resulted in a powerful synergy for a number of other antibiotics [10].

According to the literature data, the mechanism of action of essential oil components (such as terpenoids and APABs) is not the simple addition of the effects of Lev and the adjuvant, but the synergy effect that occurs [10,25,26,48]. The proposed mechanism of action of adjuvants is based on several aspects: (i) formation of bacterial membrane defects by an aromatic molecule due to its ability to interfere with the synthesis of the cell wall and cytoplasmic membrane, eventually leading to a leakage of intracellular material and, as a consequence, to the increased penetration of the antibiotics and antimicrobials (fluconazole, amphotericin, azithromycin, ciprofloxacin chloramphenicol) into the cell [10,44]; (ii) inhibition of bacterial efflux pumps resulting in an increase in the intracellular drug concentration reported previously for doxorubicin [49]; (iii) inhibition of bacterial enzymes, including dehydrogenases; (iv) suppression of the production of bacterial virulence factors, such as violacein, elastase, and pyocyanin, that prevents biofilm formation [50]. Importantly, similarly to eukaryotic cells, bacteria have multidrug transporter proteins responsible for transmembrane drug efflux and multidrug resistance development [51]. In this respect, it should be noted that apiol was reported to exhibit synergistic cytotoxic effects with doxorubicin and vincristine associated with blocking the P-glycoprotein transmembrane efflux pump [52].

In summary, the experiments have demonstrated a significant potential of APAB-CD and terpenoid-CD complexes to enhance the antibacterial effect of Lev. The results can be considered promising for the further design of novel drug formulations with improved characteristics, such as low toxicity and increased therapeutic effect.

## 3. Materials and Methods

### 3.1. Chemicals

Methyl-β-cyclodextrin (MCD) and 2-hydroxipropyl-β-cyclodextrin (HPCD) were obtained from Sigma Aldrich (St. Louis, MI, USA), levofloxacin (Lev) from Zhejiang Kangyu Pharm Co Ltd. (Zhejiang, China). Other chemicals: organic solvents, salts and acids—production Reakhim (Moscow, Russia). Menthol and linalool were purchased from Rotichrom GC (Carl Roth GmbH + Co. Karlsruhe, Germany). Eugenol and safrole at the highest commercial quality were purchased from Acros Organics (Flanders, Belgium). Components for LB medium were bacto tryptone, agarose and yeast extract (Helicon, Moscow, Russia), NaCl (Sigma Aldrich, St. Louis, MI, USA).

### 3.2. Obtaining, Isolation of APAB

Isolation of plant APAB: liquid CO_2_ extraction of parsley and dill seeds was carried out earlier by Company Karawan Ltd. (Krasnodar, Russia). APAB (apiol, dillapiol, Myristicin, allyltetramethoxybenzene) with 98–99% purity were obtained by high-efficiency distillation using a pilot plant device at N.D. Zelinsky Institute of Organic Chemistry RAS (Moscow, Russia) [53,54,55]. The seed essential oils of parsley varieties cultivated in Russia contained 70–75% of apiol (var. Sakharnaya), 21% of allyltetramethoxybenzene (var. Slavyanovskaya), and 40–46% of myristicin (var. Astra). Indian dill seeds were purchased from Vremya & Co. (St. Petersburg, Russia). The dill seeds’ essential oil contained 30–33% of dillapiol [41].

### 3.3. Preparation of β-Cyclodextrin Inclusion Complexes

(1) EG and apiol inclusion complexes with HPCD were obtained by mixing a drop of oil (1–2 mg, apiol and menthol were melted at 40 °C) with a solution of HPCD (0.23 mM) or MCD (50 and 100 mg/mL) to a total volume of 0.4 mL. The molar excess of CD varied from 0.04 to 10). Further, the complexes were incubated for 1 h at 37 °C and then centrifuged (rpm, 15 min, Eppendorf tubes) to separate the insoluble fractions. Complexes were studied by UV and FTIR spectroscopy. MCD was selected for experiments on bacterial cells due to higher LC and EE parameters.

(2) Lev, EG, apiol, dillapiol, myristicin, allyltetramethoxybenzene, linalool, menthol and safrole inclusion complexes with MCD (molar ratio 1:2) were prepared by mixing corresponding samples (approximately 2–5 mg per aromatic substance and 25–50 mg per MCD) followed by addition of 50 µL of acetonitrile. Then the mixture was intensively rubbed (kneading method), and if necessary, 25 µL of acetonitrile was added again and the procedure was repeated. Finally, incubation for 1 h at 37 °C was performed.

### 3.4. UV Spectroscopy

UV spectra of solutions (point 3.3.1) were recorded on the AmerSham Biosciences UltraSpec 2100 pro device (USA) three times in the range of 200–400 nm in a quartz cell Hellma 100–QS with an optical path of 1 cm. Background spectrum (5 mM sodium phosphate buffer (pH 6.2) with 50% EtOH) was subtracted as a blank. Calibration dependences are shown in Figure 1C.

### 3.5. FTIR Spectroscopy

ATR-FTIR spectra of samples’ solutions (point 3.3.1) were recorded using a Bruker Tensor 27 spectrometer equipped with a liquid nitrogen cooled MCT (mercury cadmium telluride) detector. Samples were placed in a thermostatic cell BioATR-II with ZnSe ATR element (Bruker, Ettlingen, Germany). The IR spectrometer was merged with a constant flow of N_2_ (NiGen HF-1, Clint, Italy). FTIR spectra were acquired from 900 to 3000 cm^−1^ with 1 cm^−1^ spectral resolution. For each spectrum, 50–70 scans were accumulated at 20 kHz scanning speed and averaged. ATR-FTIR spectra of solid samples placed on KBr glass were recorded using a Bruker Lumos II IR microscope in the region from 700 to 4000 cm^−1^ with 1 cm^−1^ spectral resolution with scanning in the area on average 1 × 1 microns. Spectral data were processed using the Bruker software system Opus 8.2.28 (Bruker, Germany), which includes linear blank subtraction, baseline correction, differentiation (second order, 17–21 smoothing points), min–max normalization and atmosphere compensation [29,56]. If necessary, 5-point Savitsky–Golay smoothing was used to remove noise. Peaks were identified by standard Bruker picking-peak procedure.

### 3.6. Antibacterial Activity of Lev and Adjuvants

The strains used in this study were *Escherichia coli* (NCIB 12210) and *Bacillus subtilis* (NCIB 8054) from National Resource Center All-Russian collection of industrial microorganisms SIC “Kurchatov Institute”). The culture was cultivated for 18–20 h at 37 °C to CFU ≈ 5 × 10^6^ – 8 × 10^6^ (determined by A_600_) in liquid nutrient medium Luria–Bertani (pH 7.2): *E. coli*—without stirring, *B. subtilis*—with stirring 180 rpm.

The study of the antibacterial effect of Lev and adjuvants was carried out by (a) agar diffusion test and (b) broth micro-dilution method. (a) A measured 500 µL of 100-fold culture dilution was evenly distributed over a solid nutrient medium (LB, pH 7.2) on a Petri dish. After 20 min, agar disks with a diameter of 9 mm were removed from the cups and 50 µL of the test samples were placed in the formed wells at the required concentration in a sterile PBS buffer (see Section 3.3 (2)). After 30 min, Petri dishes were placed in a 37 °C thermostat. After 24 h, the diameters of the growth inhibition zones were measured. Minimum inhibitory concentrations (MICs) are determined as the infimum of multiple concentrations of an antimicrobial that inhibits the visible growth of a microorganism after overnight incubation. (b) According to the standard broth micro-dilution method, 200 µL of the overnight culture (5 × 10^6^ CFU) was placed into a 96 micro-titer plate wells. Then 50 µL of tested samples was added. The range of tested Lev concentration was 4–200 ng/mL, adjuvants concentration was in the range of 0.008–2 mg/mL. Optical density measurements at 600 nm were taken at various times (0, 3, and 18 h) to determine the number of cells. The dependence of CFU (determined by A_600_) on the concentration of Lev, APAB or terpenoid (briefly X) was approximated according to the Boltzmann equation (Origin software): CFU = (control CFU)/(1 + exp(([X] − MIC_50_)/constant value)). MIC_50_ and MIC_90_ are defined as concentrations at which the inhibition of bacterial growth by 50 and 90% is noticeable, respectively. In other words, the concentration at which the ordinate value (CFU) is equal to 50 or 10% of the control sample minus the background.

### 3.7. Mathematical Calculations and Equations

(1) Calculation of the dissociation constants X–MCD and X–HPCD, where X is a “guest” compound, considering the equilibrium: X_aq_ + n MCD_aq_ ↔ X·nMCD_aq_, where *K*_d_ = [MCD_aq_]^n^ · [X_aq_]/[X·nMCD_aq_].

First, fitting of the curves of change of the analytical signal ξ (peak intensity or peak position) versus concentration of the MCD or HPCD or was carried out using the Hill equations: (1) ξ = ξ_∞_ · [MCD]^n^/([MCD]^n^ + *K*^n^), where ξ_∞_—horizontal asymptote; (2) ξ = ξ_0_ + (ξ_∞_–ξ_0_) · [MCD]^n^/([MCD]^n^ + *K*^n^). Second, calculation n and *K*_d_ values by Hill’s linearization in n-binding site model: lg (θ/(1 – θ)) = n · lg [L] – lg *K*_d_, where θ = |(ξ – ξ_0_)/(ξ_∞_ − ξ_0_)|is a fraction of the bound substance. N in Hill equation—number of drug molecules per CD torus.

Calculations for the following items are performed when [MCD] = 0.01 M.

(2) Entrapment efficiency of X [24]: EE (%) = 100 · (X amount in form of inclusion complex with CD)/(total X amount) = 100 · [X·nMCD_aq_]/([X_aq_] + [X·nMCD_aq_]) = 100 · [MCD]^n^/(*K*_d_ + [MCD]^n^).

(3) Solubility of X in water C_0_ = [X_aq_] = A/(ε · l) was determined by UV absorbance after centrifugation of samples (10 min, 10,000× *g*). l = 1 cm. ε—molar absorbance coefficient (Figure 1C).

(4) Solubility of X in water in the presence of MCD: C_max_ = [X_aq_] + [X · nMCD_aq_] = C(MCD)·EE/n + C_0_, where C(MCD) = 0.002 or 0.01 M.

(5) Loading capacity of X [14]: LC(%) = 100·(mass of loaded X)/(mass of sample) = 100 · (number of drug molecules per MCD torus) · (molar mass of X)/(molar mass of MCD); molar mass of MCD = 1310 g/mol.

(6) The effectiveness of cooperative antibacterial action ϕ of the adjuvant and levofloxacin: ϕ_adjuvant_ = (D/D_adjuvant_)^2^, ϕ_Lev_ = (D/D_Lev_)^2^, where D_adjuvant_, D_Lev_ and D—are diameters of bacterial inhibition zone caused by only adjuvant–MCD, only Lev–MCD and by Lev–MCD enhanced with adjuvant (APAB or terpenoid) in MCD’s complex, respectively. 

(7) In the context of evaluating the activity of the drug associations, the sum of fractional inhibitory concentrations was calculated from the following equation based on [25,47]:$$\mathrm{FICI}=\frac{\left[\mathrm{Lev}\right]}{{\mathrm{MIC}}_{\left(\mathrm{Lev}\right)}}\times \frac{\left[\mathrm{adjuvant}\right]}{{\mathrm{MIC}}_{\left(\mathrm{adjuvant}\right)}}$$
$$\mathrm{average}\;\mathrm{FICI}=\frac{1}{n}\overset{n}{\underset{i=1}{\sum }}\mathrm{FICI}$$
where [Lev]—fixed Lev concentration, [adjuvant]—the FIC of adjuvant, the minimum fractional concentration at which the antibacterial effect is still manifest: FIC_adjuvant_ = (MIC_adjuvant combined with Lev_)/(MIC_adjuvant alone_). 

The FICI was interpreted as: synergism (<0.35), additivity (0.35–0.75), indifference (0.76–1.4), or antagonism (>1.4). The essence of this expression is to calculate the average change in the effective concentration compared to MIC. Note that the most important parameter corresponds to a decrease in MIC of Lev. However, FICI is designed to determine the type of interaction Lev with adjuvants.

(8) Statistical analysis of obtained data was carried out using the Student’s *t*-test Origin 2022 software (OriginLab Corporation). Values are presented as the mean ± SD of three experiments.

(9) The diameters of the hydrophobic cavities CD, MD and HPCD, as well as the heights of the rounds were calculated using a computer simulation in the Avogadro program (https://avogadro.cc/ accessed on 1 July 2022) using the GAFF force field, the steepest descent (n = 2000).

### 3.8. NMR Spectroscopy

The ^1^H NMR spectra of the MCD and HPCD inclusion complexes with EG (2:1), apiol (2:1) and mixture of Lev and EG (4:1:1) complexes were recorded on a Bruker AM 400 (one-dimensional) or on a Bruker AV600 with following parameters: SF = 600.13 MHz, SI = 16 K, SW = 12,019, O1 = 2820, PW = 7.1, AQ = 1.363, RD = 5.00, NS = 4, SR = 3.91, T = 295 K with D_2_O as internal standard.

## 4. Conclusions

The formation of non-covalent guest–host complexes with CDs was previously known as a way to increase the solubility of poorly soluble organic molecules in an aqueous medium. Here, we have successfully applied this approach to a promising class of organic molecules—potential antibiotic enhancers, namely, EG, apiol, dillapiol, myristicin, allyltetramethoxybenzene, linalool, menthol and safrole. 

In the presence of a molar excess of CD (up to 5–10 fold), it was possible to achieve complete dissolution of APABs and terpenoids in an aqueous medium (at 80–98% encapsulation) higher by 10^1^–10^3^ times. The molecular details of the observed interaction were established using FTIR, ^1^H MNR and UV spectroscopy. It turned out that CD and APAB form supramolecular structures, where the most hydrophobic aromatic and allyl parts of the APAB molecule are located in the depth of the hydrophobic cavity of the CD, whereas the most hydrophilic OH group actively interacts with the external environment. Antimicrobial terpenoids and APABs were used as adjuvants capable of modifying antibiotic potential. 

The structure–activity relationship study of adjuvants with different numbers and positions of benzene ring substituents identified the optimal structure of synergic (or enhancing) molecules, namely, allyl-3-methoxy-4-hydroxybenzene, EG, which decreases Lev MIC by 3–4 times. Safrole with methylenedioxy moiety showed a somewhat weaker effect.

A synergistic antibacterial effect of APABs with Lev was detected when the Lev concentration was equal to MIC or even less. This effect was demonstrated on both Gram-positive and Gram-negative bacteria, suggesting a complex mechanism of the observed synergy that requires further elucidation. 

The observed synergy between Lev and the adjuvants is the main outcome of the study. Specifically, the two–three-fold increase in Lev efficacy caused by EG and menthol in inclusion complexes with MCD seems to be very promising in terms of reducing the dosage of toxic drugs, not only to increase the antimicrobial therapeutic effect but also for convenience in drug formulation handling. A deeper understanding of the mechanisms of these synergistic effects could facilitate further design of more effective and safe adjuvants for both existing and developed drugs.

## Figures and Tables

**Figure 1 pharmaceuticals-15-00861-f001:**
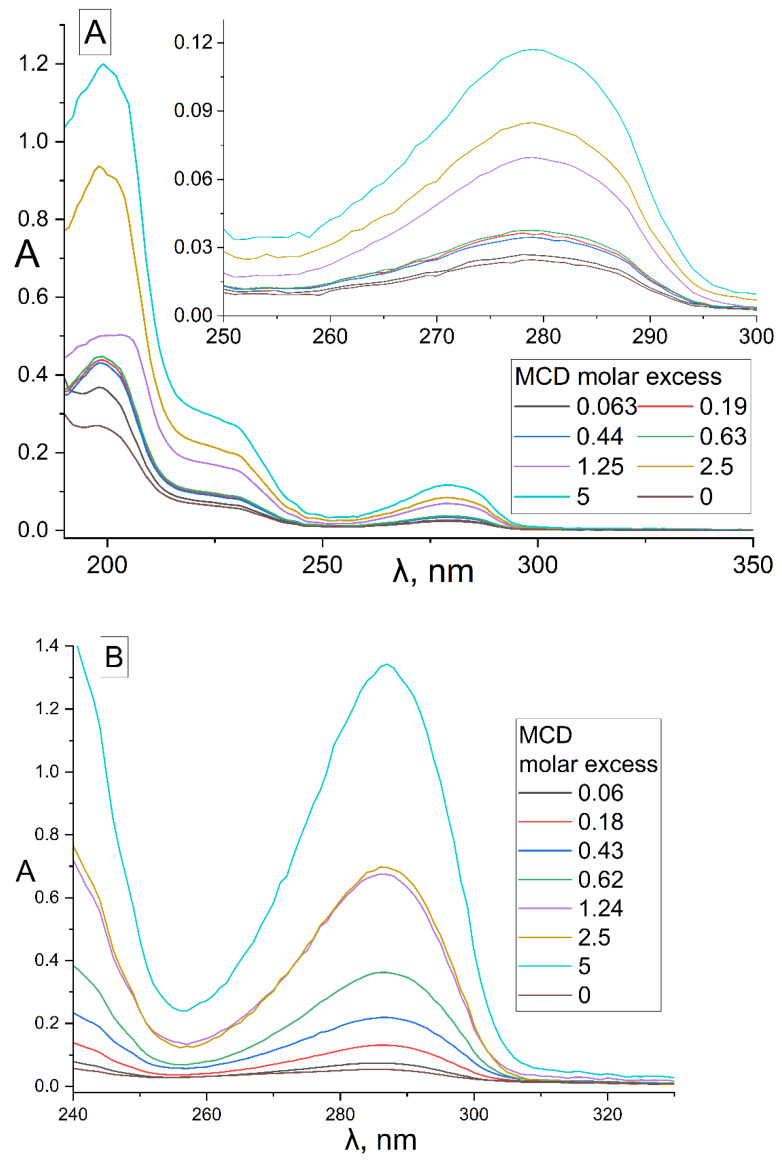
UV spectra of (**A**) EG, 15 μM, (**B**) safrole, 385 μM, and their inclusion complexes with MCD. (**C**) Calibration graphs of UV absorbance against concentration of terpenoids and APABs in 5 mM sodium phosphate buffer (pH 6.2) with 50% EtOH at 22 °C. Conditions are given in Section 3, Section 3.3 and Section 3.4.

**Figure 2 pharmaceuticals-15-00861-f002:**
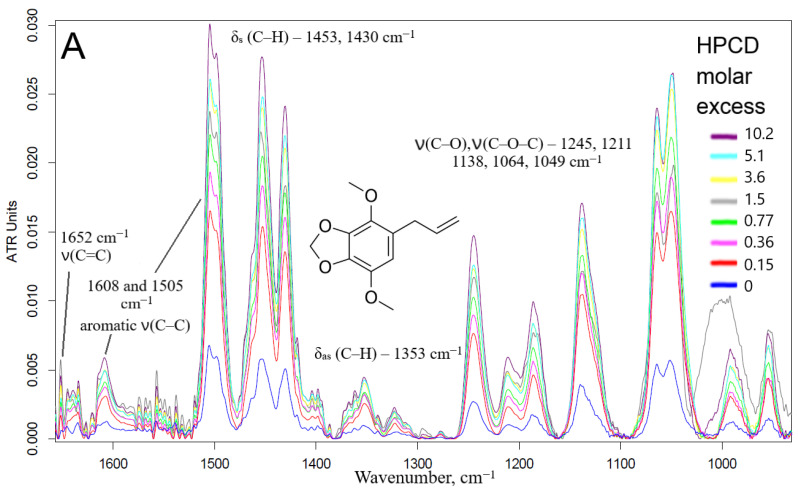
(**A**) FTIR spectra of apiol (15.8 mM) and its inclusion complexes with HPCD. (**B**) Second derivative (d^2^A/dν^2^) of FTIR spectra from (**A**). (**C**) FTIR spectra of EG, 15 mM and its inclusion complexes with HPCD. Conditions and mathematical calculations are given in Methods section (Section 3.5 and Section 3.7). Spectra of HPCD in the corresponding concentration were subtracted as background.

**Figure 3 pharmaceuticals-15-00861-f003:**
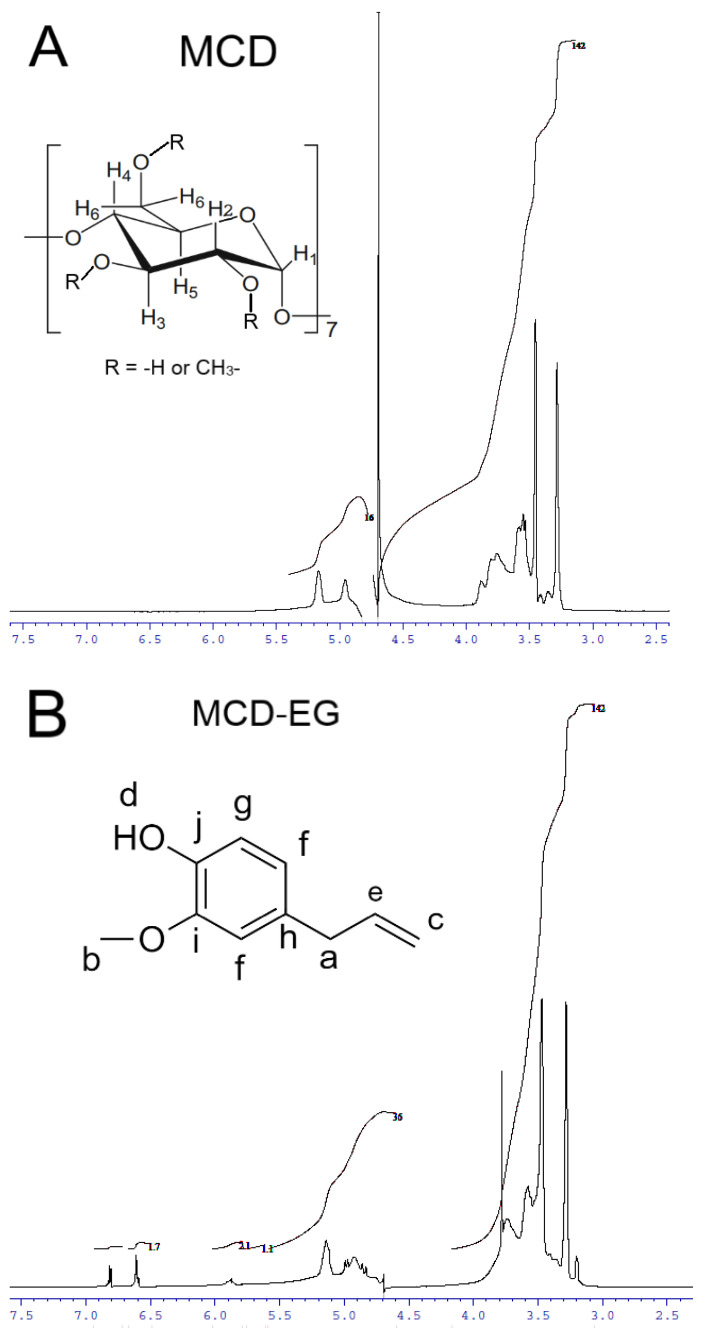
^1^H NMR-spectra (D_2_O, 600 MHz) of MCD (**A**), MCD-EG, molar ratio 2:1 (**B**), MCD-Lev-EG, molar ratio 4:1:1 (**C**). (**D**) Proposed structure of MCD inclusion complexes with EG, apiol, Lev. ^1^H NMR-spectra (D_2_O, 500 MHz) of HPCD (**E**), HPCD-apiol, molar ratio 2:1 (**F**).

**Figure 4 pharmaceuticals-15-00861-f004:**
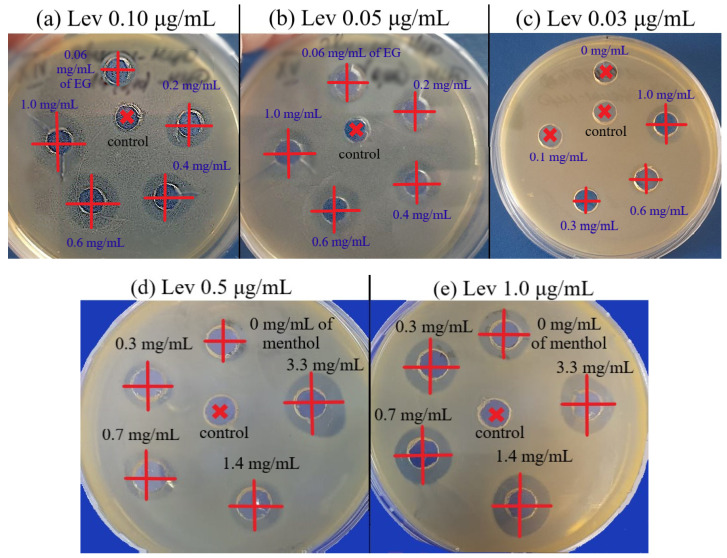
*E. coli* growth inhibition by Lev–MCD (1:2), C_Lev_ = 0.10 (**a**); 0.05 (**b**) and 0.03 μg/mL (**c**) with adjuvant EG–MCD (1:2, the concentrations of EG are indicated in the figure). *B. subtilis* growth inhibition by Lev–MCD (1:2), C_Lev_ = 0.5 (**d**) and 1.0 (**e**) μg/mL with adjuvant menthol–MCD (1:2, the concentrations of menthol are indicated in the figure). The molar ratio of the components is given in parentheses. Small bold red crosses refer to the absence of bacterial growth inhibition. Red crosses indicate the diameter of growth inhibition zone. pH 7.4 (0.01 M PBS), 37 °C, 24 h of incubation. MCD: methyl-β-cyclodextrin.

**Figure 5 pharmaceuticals-15-00861-f005:**
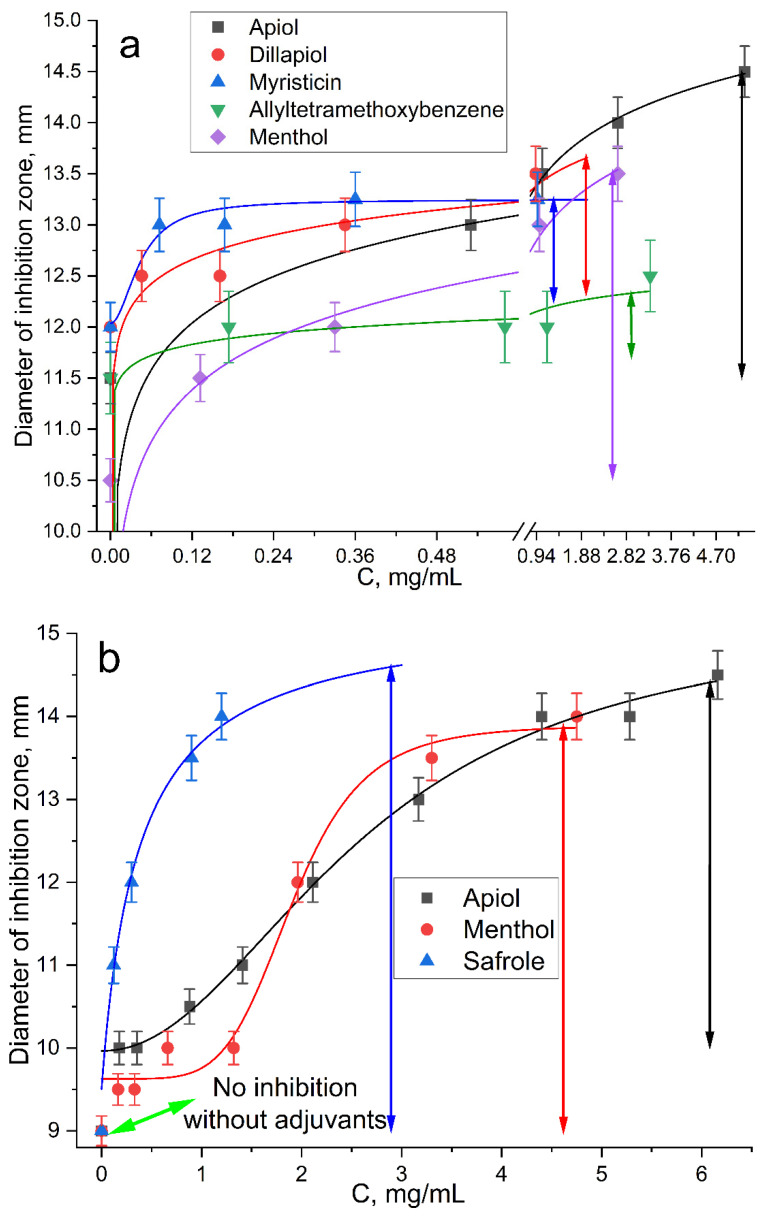
Diameter *E. coli* growth inhibition zone on Petri dishes versus concentration of adjuvant–MCD (molar ratio is 1:2) in complex formulation with (**a**) 0.10 and (**b**) 0.15 μg/mL of Lev. The diameter of the hole was 9 mm. Inhibition is considered noticeable at D > 10 mm. pH 7.4 (0.01 M PBS), 37 °C, 24 h of incubation.

**Figure 6 pharmaceuticals-15-00861-f006:**
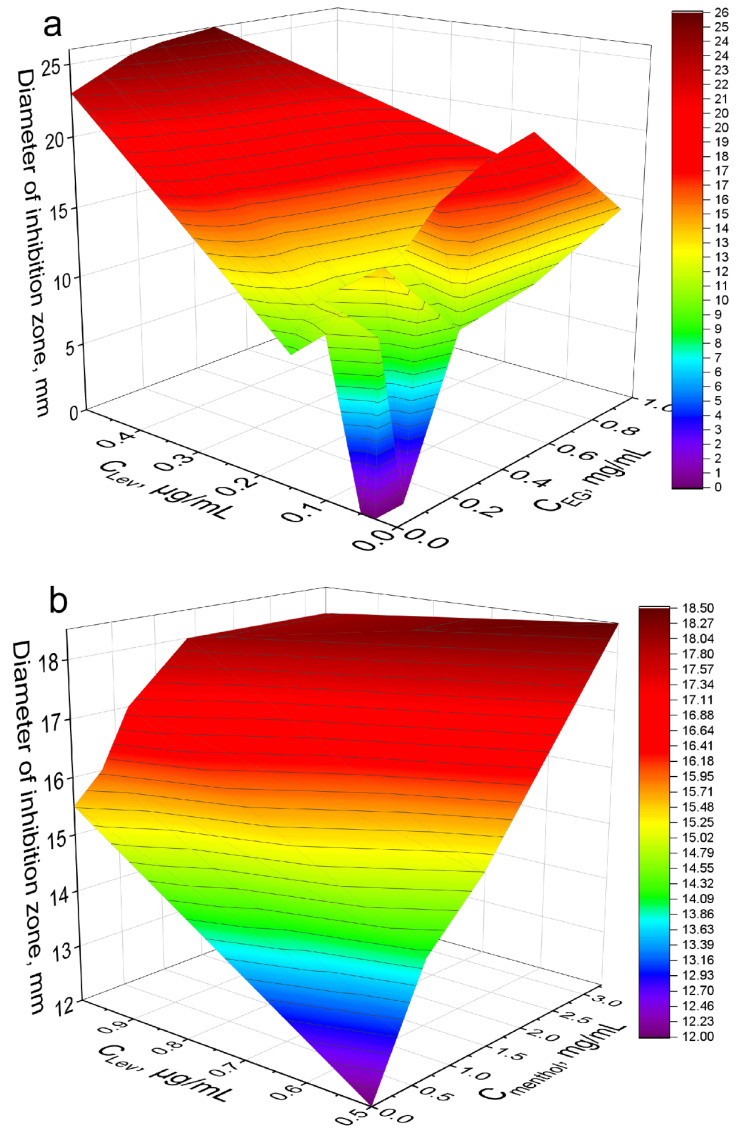
The synergy effect of EG-MCD and menthol-MCD with Lev-MCD. (**a**) Diameter of *E. coli* growth inhibition zone on Petri dishes versus concentration of EG-MCD (1:2) in complex formulation with Lev–MCD (1:2). (**b**) Diameter of *B. subtilis* growth inhibition zone versus concentration of menthol-MCD (1:2) in complex formulation with Lev–MCD (1:2). The molar ratio of the components is given in parentheses. pH 7.4 (0.01 M PBS), 37 °C, 24 h of incubation.

**Table 1 pharmaceuticals-15-00861-t001:** Physicochemical parameters of formation of MCD and HPCD inclusion complexes with Lev, APAB and terpenoids. Dissociation constants X–MCD and X–HPCD, where X is a “guest” compound, for the equilibrium: X_aq_ + n MCD_aq_ ⇌ X·nMCD_aq_. Entrapment efficiency (EE) and loading capacity (LC) are shown, C(MCD) = 0.01 M. Solubility of X in water C_0_ = [X_aq_] and solubility of X in water in the presence of 2 and 10 mM MCD − C_max_ are given. The data are indicated for MCD unless otherwise agreed. A 5 mM sodium phosphate buffer was used (pH 6.2). T = 22 °C.

Compound X	Structure of X	n	−lg *K*_d_	EE, %	LC, %	C_0_, mM	C_max_ in the Presence of 2 mM MCD/HPCD *, mM	C_max_ in the Presence of 10 mM MCD/HPCD *, mM
Levofloxacin [28]	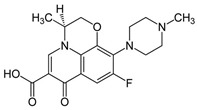	1	4.0 ± 0.5 (MCD), 3.0 ± 0.5 (HPCD)	99 ± 12 (MCD), 91 ± 15 (HPCD)	28.3 ± 0.4 (MCD), 24.1 ± 0.5 (HPCD)	67 ± 1	69 ± 1 (MCD), 68 ± 1 (HPCD)	77 ± 3 (MCD), 76 ± 4 (HPCD)
Eugenol	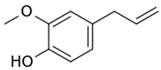	1.32 ± 0.08 (MCD), 1.2 ± 0.1 (HPCD)	2.44 ± 0.20 (MCD), 2.5 ± 0.3 (HPCD)	39 ± 3 (MCD), 56 ± 7 (HPCD)	10 ± 1 (MCD), 9 ± 1 (HPCD)	8.2 ± 0.2	8.3 ± 0.2 (MCD), 8.5 ± 0.1 (HPCD)	11 ± 1 (MCD), 13 ± 1 (HPCD)
Apiol	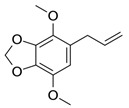	1.6 ± 0.2 (MCD), 1.56 ± 0.25 (HPCD)	2.6 ± 0.3 (MCD), 2.0 ± 0.4 (HPCD)	20 ± 2 (MCD), 7 ± 1 (HPCD)	11 ± 1 (MCD), 9 ± 1 (HPCD)	0.13 ± 0.01	1.3 ± 0.1 (MCD, HPCD)	2.6 ± 0.3 (MCD), 1.8 ± 0.3 (HPCD)
Dillapiol	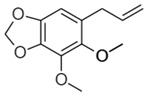	1.3 ± 0.2	2.7 ± 0.5	56 ± 10	13 ± 1	0.24 ± 0.05	0.45 ± 0.07	4.5 ± 0.7
Myristicin	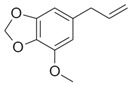	0.67 ± 0.06	3.52 ± 0.16	99 ± 5	22 ± 2	0.030 ± 0.007	3.0 ± 0.2	15 ± 1
Safrole	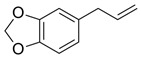	1.5 ± 0.2	4.3 ± 0.4	95 ± 4	8 ± 1	0.8 ± 0.1	1.7 ± 0.1	7.1 ± 0.9
Allyltetramethoxybenzene	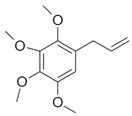	1.74 ± 0.11	3.4 ± 0.3	45 ± 4	11 ± 1	0.16 ± 0.02	0.22 ± 0.03	2.8 ± 0.4
Linalool	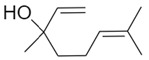	1.2 ± 0.2	3.0 ± 0.4	80 ± 7	10 ± 1	0.30 ± 0.05	0.91 ± 0.07	7 ± 1
Menthol	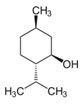	1.1 ± 0.1	3.2 ± 0.3	91 ± 6	11 ± 1	0.23 ± 0.04	1.4 ± 0.2	8.5 ± 0.8

* MCD—unless otherwise specified.

**Table 3 pharmaceuticals-15-00861-t003:** Minimum inhibitory concentrations (MIC) of Lev, APAB and terpenoids X in the form of inclusion complexes with MCD. *E. coli* (5 × 10^6^ CFU) and *B. subtilis* (7 × 10^6^ CFU). LB medium (pH 7.2), 37 °C, 20 h.

Compound	*E. coli* (NCIB 12210)			*B. subtilis* (NCIB 8054)		
	MIC *, mg/mL	MIC_50_ **, mg/mL	MIC_90_ **, mg/mL	MIC *, mg/mL	MIC_50_ **, mg/mL	MIC_90_ **, mg/mL
Lev	0.10 ± 0.01 μg/mL	0.010 ± 0.002 μg/mL	0.025 ± 0.004 μg/mL	0.45 ± 0.05 μg/mL	0.03 ± 0.01 μg/mL	0.08 ± 0.02 μg/mL
Lev-MCD	0.06 ± 0.01 μg/mL	0.007 ± 0.002 μg/mL	0.022 ± 0.003 μg/mL	0.25 ± 0.02 μg/mL	0.016 ± 0.005 μg/mL	0.05 ± 0.01 μg/mL
EG-MCD	0.30 ± 0.05	0.04 ± 0.02	0.13 ± 0.03	1.0 ± 0.1	0.07 ± 0.02	0.15 ± 0.05
Apiol-MCD	2.6 ± 0.2	1.2 ± 0.3	>2	5.3 ± 0.4	>2	
Dillapiol-MCD	2.4 ± 0.3	1.0 ± 0.1	>2	>5	>2	
Myristicin-MCD	>3	1.9 ± 0.5	>2	>5	0.20 ± 0.04	>2
Safrole-MCD	1 ± 0.1	0.15 ± 0.04	0.8 ± 0.1	3.9 ± 0.3	0.10 ± 0.01	>2
Allyltetramethoxybenzene-MCD	>3	>2		>5	>2	
Linalool-MCD	>3	>2		>5	>2	
Menthol-MCD	>3	>2		2.6 ± 0.3	0.15 ± 0.03	>2

* Agar diffusion test; ** broth micro-dilution method.

**Table 4 pharmaceuticals-15-00861-t004:** Parameters of *E. coli* and *B. subtilis* growth inhibition by mixtures of Lev-MCD and adjuvant-MCD (1:2). D_Lev_ and D—diameter of bacterial inhibition zone surrounding Lev–MCD and Lev–MCD with APAB-MCD, respectively. ϕ_Lev_—cooperative antibacterial action of APABs or terpenoids and Lev. The conditions and adjuvant concentrations are similar to those given in Table 3. Values are presented as the mean ± SD of three experiments.

Compound X-MCD	C_Lev_, μg/mL	D_Lev_, mm (±0.5)	D, mm (±0.5)	ϕ_Lev_ = (D/D_Lev_)^2^ (±0.1)	FIC_adjuvant_ *, mg/mL	FICI ** (±15%)	Type of Interaction between Lev and X
*E. coli*							
Eugenol	0.03	NI	14	2.0	0.30 ± 0.005	0.30	Synergism
	0.05	NI	17	2.9	0.05 ± 0.01	0.08	
	0.1	11	20	3.3	0.030 ± 0.005	0.10	
	0.15	10	15	2.3	0.010 ± 0.002	0.05	
	0.5	22.5	26	1.3	0.08 ± 0.01	1.3	Indifference
Apiol	0.1	11.5	14.5	1.6	0.9 ± 0.1	0.35	Additivity
	0.15	NI	14	2.0	0.35 ± 0.05	0.20	Synergism
Dillapiol	0.1	12	13.75	1.3	0.115 ± 0.015	0.04	
Myristicin	0.1	12	13.25	1.2	0.12 ± 0.02	0.04	
Allyltetramethoxybenzene	0.1	11.5	12	1.1	2.0 ± 0.2	0.50	Additivity
Linalool	0.1	NI	NI ***	NI	NI	NI	Indifference
Menthol	0.1	10.5	13.5	1.7	0.06 ± 0.01	0.02	Synergism
	0.15	NI	14	2.0	0.40 ± 0.05	0.15	
Safrole	0.15	10	14.5	2.1	0.06 ± 0.01	0.09	
*B. subtilis*							
Eugenol	0.4	12	14.5	1.5	0.10 ± 0.01	0.16	Synergism
	0.7	14.5	16	1.2	0.2 ± 0.02	0.56	Additivity
	1.5	17	19	1.2	0.10 ± 0.01	0.60	
Safrole	0.4	12	14	1.4	0.17 ± 0.02	0.08	Synergism
	0.7	15	16	1.1	0.33 ± 0.03	0.28	
	1.5	17	20	1.4	0.15 ± 0.02	0.27	
Menthol	0.5	12	18.5	2.4	0.17 ± 0.01	0.11	
	1	15.5	18	1.3	0.17 ± 0.01	0.23	

* FIC—the minimum fractional concentration at which the antibacterial effect is still manifest; FIC_adjuvant_ = (MIC_adjuvant combined with Lev_)/(MIC_adjuvant alone_); ** $\;\mathrm{FICI}=\frac{\left[\mathrm{Lev}\right]}{{\mathrm{MIC}}_{\left(\mathrm{Lev}\right)}}\times \frac{\left[\mathrm{adjuvant}\right]}{{\mathrm{MIC}}_{\left(\mathrm{adjuvant}\right)}}$; *** NI—no inhibition.

## Data Availability

The data presented in this study are available in the main text and Appendix A.

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
