# Peer review of "Plant Alkylbenzenes and Terpenoids in the Form of Cyclodextrin Inclusion Complexes as Antibacterial Agents and Levofloxacin Synergists"

_pharmaceuticals, 2022, doi:10.3390/ph15070861_

Round 1

Reviewer 1 Report

The manuscript is very interesting, but if the objective is to increase the solubility of these compounds in water, I consider that the most appropriate test to determine the MIC is in a liquid culture medium.

I suggest that the authors perform the broth dilution micro technique assay and compare their results with those shown in the manuscript.

Author Response

Dear reviewers!

The authors of this paper sincerely thank you for studying the article and writing a constructive review! All comments have been taken into account and appropriate improvements/corrections/experiments have been carried out. The following is a description of the changes in the work that we made during the revision.

Reviewer's comments: «The manuscript is very interesting, but if the objective is to increase the solubility of these compounds in water, I consider that the most appropriate test to determine the MIC is in a liquid culture medium. I suggest that the authors perform the broth dilution micro technique assay and compare their results with those shown in the manuscript».

Authors’ response:

Thank you for this comment.

Authors fully agree with Reviewer's comments: Indeed, the comparison of the results of MIC determining by the Agar Diffusion test with those obtained by broth micro-dilution method is important to verify the trends. So, we conducted the recommended by the reviewer experiments and calculated the values of MIC50 and MIC90 for Lev, Lev–MCD and adjuvants–MCD (Table 3 in MS). Trends in the values relative characteristics for the studied set of APABs and terpenoids complexes correlate with those obtained by the diffusion in agar test. However, it should be noted that the presented values of MIC50 and MIC90 are less than in terms of diffusion in agar due to differences in the experiment conditions: slow diffusion rate a in agar and a large bacterial seeding area. The values obtained by the two methods correlate with each other in terms of relative characteristics. Comparison of adjuvants MIC values yielded EG as optimum molecular scaffold, namely, allylbenzene with 3-methoxy-4-hydroxy substituents. Safrole with methylenedioxy moiety was the second in activity range, whereas apiol and dillapiol featuring dimethoxy-methylenedioxy fragment were less potent (Table 3). Thus, as found by both Agar Diffusion test and by broth micro-dilution method Structure-activity relationship for adjuvants on gram-positive and gram-negative bacteria is similar (Table 3). A ranked list of substances according to their activity against E. coli: Lev-MCD > Lev >>> EG-MCD > dillapiol-MCD ≈ apiol-MCD > myristicin-MCD >> other. A ranked list of substances according to their activity against B. subtilis: Lev-MCD > Lev >>> EG-MCD > safrole-MCD ≈ menthol-MCD > myristicin-MCD >> other.

Table 3

Compound

E.coli

(NCIB 12210)

B.subtilis (NCIB 8054)

MIC*, mg/mL

MIC50**, mg/mL

MIC90**, mg/mL

MIC*, mg/mL

MIC50**, mg/mL

MIC90**, mg/mL

Lev

0.10±0.01 μg/mL

0.010±0.002 μg/mL

0.025±0.004 μg/mL

0.45±0.05 μg/mL

0.03±0.01 μg/mL

0.08±0.02 μg/mL

Lev-MCD

0.06±0.01 μg/mL

0.007±0.002 μg/mL

0.022±0.003 μg/mL

0.25±0.02 μg/mL

0.016±0.005 μg/mL

0.05±0.01 μg/mL

EG-MCD

0.30±0.05

0.04±0.02

0.13±0.03

1.0±0.1

0.07±0.02

0.15±0.05

Apiol-MCD

2.6±0.2

1.2±0.3

>2

5.3±0.4

>2

Dillapiol-MCD

2.4±0.3

1.0±0.1

>2

>5

>2

Myristicin-MCD

>3

1.9±0.5

>2

>5

0.20±0.04

>2

Safrole-MCD

1±0.1

0.15±0.04

0.8±0.1

3.9±0.3

0.10±0.01

>2

Allyltetramethoxybenzene-MCD

>3

>2

>5

>2

Linalool-MCD

>3

>2

>5

>2

Menthol-MCD

>3

>2

2.6±0.3

0.15±0.03

>2

The authors are grateful to the reviewer for valuable comments.

Reviewer 2 Report

The author described the formation of an inclusion complex between the terpenoids and cyclodextrin (CDs) to enhance the solubility of bioactive substances. They used IR and UV/Vis methods to determine the formation of IC. However, the estimation of IC is chaotically written and contains many miss-understanding. Although, even though this concept has been used a lot in the previous works, this study can be published after major revisions addressing the following points:

The author studied the formation of inclusion complex between the Eu or APAB and HPCD, however, for the microbiological assay they used another terpenoid without determining of ability to the formation of inclusion complex. Additionally, they studied the antibacterial properties of the MCD inclusion complex, not HPCD. It is not clear to me and I think also to the readers why the authors change the CD. I think that the authors should unify it. For this purpose, the author should investigate the formation of IC between MCD and all terpenoids. 

Did the authors try to elucidate the structure of IC?  It means, which part of terpenoids is entrapped inside the CD’s cavity? I recommend using the 2D NMR or IC-MS method described in the literature (Materials Today Communications 25 (2020) 101605 or Carbohydrate Polymers 209, (2019), 74-81).

The authors claimed that there are no differences between the HPCD and MCD. However, it is known that the cavity diameter depends on the used substituents. The author should, first, perform the X-ray of both CDs and check the differences between diameters. (A.-W. Cheng, J.-P. Wang, Z. Jin, Preparation and analysis of cyclodextrin. Cyclodext. Chem.,  2013, pp. 83–99).

Figure 2, please remove the Brucker name.

The author calculated the dissociation constant (4.1). However, I did not find the calculation or description of how they did it. 

It is known that after the inclusion of molecules inside the CD's cavity, the intensity of peaks belonging to entrapped compounds decreases. Why did the authors observe the increase of peak intensity in the case of APAB? This observation suggests that the inclusion complex is not formed. Please verifier it. 

Please, use the ratio of CD to terpenoids instead of the concertation of HPCD in all Figures.

How many molecules can be entrapped inside the HPCD cavity? 

Round 2

Reviewer 1 Report

The authors complied with all recommendations.

Author Response

Thank you.

Reviewer 2 Report

The authors corrected the manuscript according to my suggestion thus, I recommend this article for publication. 

Author Response

Thank you.

This manuscript is a resubmission of an earlier submission. The following is a list of the peer review reports and author responses from that submission.

Round 1

Reviewer 1 Report

The manuscript is very interesting, but the following should be clarified:

The authors should clarify: did they work with terpenes? or with the essential oil? If they worked with the essential oil, they must include the chemical composition of each sample.

Line 2.  Essential oils? Or terpenoids? Clarify.

Line 12. Eugenol, apiol… are terpenoids, they are not essential oils.

Line 122, 265, 273, 308, 319, 327, 332, 340, 341, 343, 350. Scientifics names in italics.

Line 139. Fig 1b is apiol change for Fig 1a.

Line 142. There is not Fig 1c.

Line 141-142. Please clarify the calculation of Cmax, according to the equation in Fig S1.

Fig. 1. Mark Fig. 1a and Fig. 1b.

Line 238. “Disolution of essential oils” change by dissolution of monoterpenes.

Throughout the manuscript is confusion between essential oil and monoterpenes, it mus be corrected.

Line 284. “and essential oils…” shoul say and monoterpenes.

Line 291, 298, 329 essential oil change by monoterpenes.

Line 428. Please clarify the bacterial density of the cell culture (how CFU/ml?).

In 3.7 Mathematical calculations and equations, please include the Cmax (EG) and Cmax (apiol) mathematical calculations.

Line 434. If the objective of this research was to increase the solubility of terpenes in an aqueous medium, why did the authors not determine the MIC in broth?

Reviewer 2 Report

The title of the manuscript does not reflect the content of the work, which was desined to prepare guest-host inclusion complexes with ß-cyclodextrin, as and then study their antibacterial activity.

Moreover: the use of the term „essential oils” is this work is not correct because authors tested isolated individulal compounds not essential oils.

The abstract is not clear: it is stated that levofloxacin-cyclodextrin complex yielded a reduced MIC value against E.coli, whereas the title of the work suggests the synergistic effect between essential oils and antibiotic.

Line 27, was the introduction of 2-3.5 mg/ml of the adjuvant menthol in the form of cyclodextrin complex?

Does soluble form mean cyclodextrin complex?

Can you explain what is combined enhanced fluoroquinolone drugs? Are these cyclodextrin complexes?

Do you have any premise that volatile compounds cause the inhibition of efflux by proteins?

Methods

The origin of eugenol and Safrole was not provided.

Line 390 add „water solution”

The method of the preparation of the guest-cyclodextrin complex should be better described and referenced. Was it kneading method? From the description it is not clear

It is not explained why eugenol and apiol inclusion complexes were prepared with HPCD, while later Levofloxacin, eugenol, apiol, dillapiole, myristicin, allyltetramethoxybenzene, linalool, menthol and safrole inclusion complexes were done with MCD?

Lines 393-398, what was the concentration of MCD? What was the method of preparation. Was MCD added in acetonitrile instead of water? The description is not clear.

Results

Lines 121-129 – the justification of the undertaken research should be placed in the introduction section

Why the detailed characteristics of the obtained complexes were shown for HPCD complexes whereas antibacterial activity was studied for MCD, and the characteristics for the latter was not given?

Figure 1 caption is not clear: The UV spectrum presented is of complexes with cyclodextrin, whereas the caption suggests that it is spectrum of apiol/eugenol and spectrum of complexes. This should be clarified. The info from line 139: “In Fig. 1b, the blue spectrum corresponds to eugenol dissolved in water without CD” Should be placed under the Figure. The same remark for Figure 2.

Table 1, Why parameters provided in the table were shown only for eugenol and apiol?

Table 2. In the caption the CFU is provided for E.coli but not for B.subtilis.

Why the activity of the free volatile compounds was not studied to compare with their activity in cyclodextrin complexes?

Why some compounds were tested in the concentration up to 5.8 mg/ml whereas other only up to 1.2 mg/ml and differently on both strains?

Table 3. The concentration of the adjuvant was 2-4 mg/ml – which compounds were 2 mg/ml, which 3 mg/ml or 4mg/ml? It is not clear. Additionally, a line above it is stated that concentration of adjuvants were 1-4 mg/ml.

Could you please provide the reference for synergism Ï• equation or explain it? The ranges of the values cosidered as synergism or neutral or antgonism should be provided.

Figure 3. The range of concentrations of eugenol added as adjuvant is different than stated in Table 2 and in the text (line 293).

There is no discussion section in the manuscript, probably authors combined results with discussion but didn’t indicate it.

Reviewer 3 Report

Regarding the MS entitled "Essential oils as antibacterial agents and levofloxacin synergists" I am not sure about its suitability to be considered for publication in Pharmaceuticals after revisions. In my opinion the aim is blur; a first glance shows that the research is not designed well. Pure volatile compounds are selected randomly, the microorganisms and antibiotic as well. The results are not significant, eugenol has also been studied previously based on the authors. Moreover, the results has not been well-discussed.